# Research on Switching Current Model of GaN HEMT Based on Neural Network

**DOI:** 10.3390/mi16080915

**Published:** 2025-08-07

**Authors:** Xiang Wang, Zhihui Zhao, Huikai Chen, Xueqi Sun, Shulong Wang, Guohao Zhang

**Affiliations:** 1School of Mechanical and Electrical Engineering, Guangdong University of Technology, Guangzhou 510006, China; garyzhang@gdut.edu.cn; 2School of Microelectronics, Xidian University, Xi’an 710071, China; 15732684598@163.com (Z.Z.); 18981894752@163.com (H.C.); 22111213950@stumail.xidian.edu.cn (X.S.)

**Keywords:** deep neural network (DNN), GaN HEMT, switching current characteristics, power devices voltage, on-resistance

## Abstract

The switching characteristics of GaN HEMT devices exhibit a very complex dynamic nonlinear behavior and multi-physics coupling characteristics, and traditional switching current models based on physical mechanisms have significant limitations. This article adopts a hybrid architecture of convolutional neural network and long short-term memory network (CNN-LSTM). In the 1D-CNN layer, the one-dimensional convolutional neural network can automatically learn and extract local transient features of time series data by sliding convolution operations on time series data through its convolution kernel, making these local transient features present a specific form in the local time window. In the double-layer LSTM layer, the neural network model captures the transient characteristics of switch current through the gating mechanism and state transfer. The hybrid architecture of the constructed model has significant advantages in accuracy, with metrics such as root mean square error (RMSE) and mean absolute error (MAE) significantly reduced, compared to traditional switch current models, solving the problem of insufficient accuracy in traditional models. The neural network model has good fitting performance at both room and high temperatures, with an average coefficient close to 1. The new neural network hybrid architecture has short running time and low computational resource consumption, meeting the needs of practical applications.

## 1. Introduction

Third-generation wide bandgap semiconductor devices, represented by gallium nitride high-electron mobility transistors (GaN HEMTs), have successfully emerged [1]. With their high-frequency switching capability (MHz GHz level), high voltage tolerance (>650 V), ultra-low on-resistance (as low as m Ω level), low dielectric constant, high power density, and high-quality quality factor, they meet the working requirements of high-power and high-frequency semiconductor devices and have become a key carrier for breaking through the boundaries of existing technology [2,3].

As a core component of a new type of semiconductor, gallium nitride high-electron mobility transistor (GaN HEMT) has been applied in high-tech fields such as 5G communication and new energy vehicles due to its wide bandgap and high electron mobility characteristics. Accurate modeling of its switching current has become a top priority [4]. However, in practical applications, the switching characteristics of GaN HEMT devices exhibit a very complex dynamic nonlinear behavior and multi-physics field coupling characteristics. There are new challenges in modeling GaN-based power devices. Firstly, high-frequency switching technology is the foundation of power electronics. The higher the switching frequency, the higher the power density and efficiency of the power exchange circuit are expected to be. The transport characteristics of two-dimensional electron gas in GaN-based HEMT structures exhibit nonlinear effects such as quantization, polarization, and non-local transport due to high voltage and high current [5]. Accurate modeling of these physical effects is necessary to describe the transport behavior of electrons under high voltage and high current. The difficulty in preparing GaN materials and devices [6] is limited by the current level of material preparation. GaN materials themselves have many defects and dislocations, and the defect models are complex. This seriously restricts the transport and mobility of charge carriers, which, in turn, has an important impact on the electrical output curve trend of the device. In addition, accurately describing the EMI/EMC and thermal simulation of GaN HEMT devices at high switching frequencies and high power in the model is also a huge challenge that urgently needs to be addressed [7].

Due to the complex physical properties and numerous physical parameters of GaN-based HEMTs, it is difficult to model them using BSIM modeling methods similar to those in CMOS, which is a physics-based model. Therefore, empirical models based on mathematical analytical expressions are more suitable for HEMT devices with complex processes and are widely used. Examples would be the Curtice model, Angelov model, and EEHEMT1 model [8,9]. At present, the models of GaN-based HEMT devices mainly consider various parasitic effects, such as thermal effects, piezoelectric effects, noise, capture effects, etc., which can reduce the performance of RF and microwave circuits. Therefore, it is necessary to improve and develop the models according to the different responses of the devices [10,11,12,13]. The current models are mainly based on the basic HEMT core model, constructing various physical models, such as the self-heating model, trap model, channel modulation model, and field plate model, which have received more attention. At the same time, traditional switch current models based on physical mechanisms have significant limitations. Due to the large number of partial differential equations and physical parameters involved in the internal physical processes of the device, complex numerical solutions are required during the simulation calculation process, which consumes huge computing resources and time. For example, the Angelov model simulates current characteristics over a wide temperature range, and the calculation time may last for several hours or even days. Moreover, due to its simplification and excessive assumptions in the calculation process, it is difficult to accurately capture the complex, nonlinear I/V characteristics of GaN HEMTs under high-frequency and large signal excitation [14], resulting in large deviations between the design of high-frequency power amplifiers and actual performance.

GaN HEMTs have complex conductive mechanisms and various non-ideal effects. In order to describe the complex physics and mathematics involved, many advanced achievements have introduced neural networks and combined them with the latest physical mechanism theories to model the devices [13]. The neural network adopts a parallel computing architecture and, with the help of GPU acceleration, can complete the traditional model’s hours-long computing tasks in minutes, greatly improving design efficiency. The complex structure composed of multiple layers of neurons has strong nonlinear mapping ability, which can be trained with a large amount of actual data to accurately fit the complex I/V nonlinear relationship of GaN HEMT and improve the accuracy of the model. The fully trained neural network has good generalization and prediction ability for new working condition data that did not appear in the training set. Faced with the complex and changing actual working environment of GaN HEMT, it can accurately predict the switching current under different working conditions, providing a reliable basis for circuit design and enhancing system stability and reliability.

At present, the use of neural networks in academic modeling of device characteristics mainly focuses on two starting points. One approach is to treat the entire period device as a black box and use a complete deep network to describe the period characteristics from the outermost periphery. The other approach still inherits the device model architecture of full physics or semi-physics and semi-experience and then uses neural networks to describe some of the mechanism modules with significantly complex rules. For the former full black box model, such as the one proposed by Mingqiang Geng, Giovanni Crupi et al., a complete model based on the LSTM network was used to model the nonlinear behavior of a 10 W RF power amplifier GaN HEMT, achieving high descriptive accuracy [15]. The latter model adopts modular thinking, such as the device model architecture based on virtual source theory proposed by An Dong Huang, Zheng Zhong, and others, which uses neural networks to model the intrinsic current source [13]. The team at the University of Science and Technology of China has developed and optimized artificial neural network algorithms to achieve high-precision modeling of GaN HEMTs over a wide temperature range (from an extremely low temperature of 4.2 K to room temperature at 300 K), with a prediction accuracy of over 99% for key performance indicators of the device. This fully demonstrates the enormous potential and role of neural network algorithms in GaN HEMT modeling [16].

Neural networks have been proven to be an algorithm with strong nonlinear function relationship description and generalization ability. Introducing neural network algorithms into the field of GaN HEMT device modeling brings a research path with optimization potential in accuracy, generalization, description efficiency, and computational speed. In terms of neural network model architecture design, this paper adopts a hybrid architecture of convolutional neural network and long short-term memory network (CNN-LSTM). In the 1D-CNN layer, the one-dimensional convolutional neural network can automatically learn and extract local transient features of time series data by sliding convolution operations on its convolution kernel on time series data, making these local transient features present a specific form in the local time window. In the double-layer LSTM layer, the neural network model captures the transient characteristics of switch current through the gating mechanism and state transfer. Through this study, it is expected to construct a more accurate and efficient switching current model, which can solve the problems of dynamic characteristic modeling, computational efficiency, and GaN HEMT characteristics under different operating conditions in existing GaN HEMT models.

While prior studies, such as [15], have successfully applied LSTM networks for behavioral modeling of RF power amplifiers, and [13,16] have utilized ANNs for electrothermal modeling, these works do not specifically address the high-fidelity prediction of the switching current waveform of GaN HEMTs. This switching process is characterized by unique and extremely fast transient phenomena, such as sharp rising/falling edges, voltage overshoots, and ringing, which pose a significant challenge for traditional modeling approaches.

The core innovation of this study lies in the synergistic application of a hybrid CNN-LSTM architecture to precisely capture these complex switching dynamics. Specifically, the 1D-CNN layer is uniquely leveraged to automatically extract local, high-frequency transient features (e.g., current spikes) from the input voltage waveforms, while the subsequent LSTM layers model the temporal dependencies and memory effects throughout the entire switching event. This combined approach provides a more robust and accurate representation of the device’s dynamic behavior compared to using a standalone LSTM or a conventional ANN. Our work demonstrates that this specific architecture is highly effective for this application, achieving excellent accuracy at both room and high temperatures, thereby providing a novel and efficient tool for power electronics circuit design and analysis.

## 2. Experimental Design

Before constructing and training the neural network model, it is necessary to collect switch current data of GaN HEMT devices. By constructing and training the neural network model on specific data, the prediction accuracy and generalization ability of the model can be better improved.

### 2.1. Pulse I–V Static DC Current Characteristics Test

Before capturing the switching current dynamic characteristics of GaN HEMT devices, static DC current characteristic testing of GaN HEMT devices must be conducted first. When conducting DC testing, even if only a short voltage excitation is applied to GaN HEMTs, self-heating and trap effects will occur, which will affect the DC current and cause changes in the DC current, interfering with the DC current data.

The self-heating effect refers to the phenomenon in which a device experiences power loss due to the current passing through its internal resistance during operation, resulting in an increase in internal temperature and further affecting its electrical characteristics [17]. The trap effect refers to the localized energy levels formed by lattice defects, surface states, or impurities in semiconductor materials. These defects and localized energy levels can capture and release charge carriers (electrons or holes), temporarily bind them, and detach them from the conductive process, thereby affecting the electrical dynamic characteristics of the device [18]. Therefore, in order to avoid the influence of the self-heating effect and trap effect on the DC characteristics of GaN HEMT devices during DC testing, a pulse I–V testing method was adopted in this paper to test the DC characteristics of the transistor [19].

Pulse I–V testing technology measures the DC characteristics of GaN HEMTs by applying short-term pulse signals to the devices. Short-term pulse signals can effectively reduce the effects of self-heating and trap effects on the DC characteristic test results. In pulse I–V testing technology, the input voltage or current signal is applied to the device in the form of pulses, with a single pulse width typically controlled in the range of hundreds of nanoseconds to microseconds. Due to the extremely short pulse duration and sufficient interval between adjacent pulses, the device can be fully cooled after each pulse action, avoiding the thermal accumulation phenomenon. This fast pulse excitation method can significantly suppress the self-heating effect caused by continuous current. By controlling the pulse width and interval reasonably, the temperature fluctuation of the device during the testing process is limited to a very small range, thereby eliminating the interference of temperature changes on the I–V characteristics.

For the trap effect, in traditional DC testing, due to the long-term exposure of the device to voltage and current, it is easy for carriers to be captured by traps, resulting in dynamic changes and unstable current and voltage characteristics. Pulse I–V testing shortens the interaction time between carriers and trap energy levels through short-term pulse signals, making the capture and release process of traps less likely to occur, avoiding the influence of trap effects, and thus obtaining stable and close-to-the-original characteristics of the device measurement data.

In the implementation process of pulse I–V testing, the main focus is on optimizing pulse parameters and setting signals. The pulse width and interval settings need to ensure that a stable DC testing platform is formed within the pulse range to avoid the accumulation of memory effects. Generally speaking, the pulse interval is several orders of magnitude longer than the pulse width to ensure sufficient heat dissipation of the device. Through such operations, this testing method can accurately obtain the I–V characteristics of devices without self-heating and trap effect interference, providing reliable data support for the performance evaluation of GaN HEMT devices.

In this experiment, the transistor pin is connected to the low-impedance probe of Keysight(Santa Rosa, CA, USA) B1505A, which can generate controlled test pulses and accurately collect the output current during the pulse period. A pulse signal with a duty cycle of 0.1% is used, where the high-level duration is 1 μs and the rise and fall times are both 100 ns. To avoid the dynamic effects of switch edges and non-ideal errors, three sampling points at positions 1/4, 2/4, and 3/4 are selected for measurement during the high-level stationary phase, and the average of the three points is taken as the final measurement data.The Pulse I–V static DC characteristics test gate voltage waveform is shown in Figure 1. The leakage voltage waveform of pulse I–V static DC characteristic test result is shown in Figure 2. And the leakage source current waveform of pulse I–V static DC characteristic test is shown in Figure 3.

### 2.2. Dynamic Current Testing of Power GaN HEMT Switches

In order to accurately measure and capture the switching dynamic current characteristics of GaN HEMT devices, this paper constructs a dual-pulse (DPT) half-bridge circuit for testing power GaN HEMT devices as shown in Figure 4. The dual-pulse test circuit is an effective tool for measuring dynamic current in switch engineering of GaN HEMT devices. The dual-pulse test is the main method for analyzing the dynamic current characteristics of power devices. The dual-pulse test precisely controls the pulse timing range of the gate signal to measure the voltage and current waveforms during transient processes. For the dynamic switching tests, a DC bus voltage of 400 V was applied across the drain and source, and the gate was driven with a pulse switching from 0 V to +6 V to trigger the transient event. These conditions were kept consistent for all dynamic measurements.

In the dual-pulse test of this article, two types of loads were designed: one was a resistive load, and the other was an inductive load [20,21]. For resistive loads, since ideal resistors do not introduce any dynamic effects and can provide stable current and voltage responses, this study chose a resistive load circuit as the excitation circuit for data acquisition and neural network model training datasets. By using resistive load circuits, it is possible to observe the switching current characteristics of GaN HEMTs more effectively without considering the influence of other dynamic effects. For inductive loads, inductive load circuits are used in standard dual-pulse testing to verify the performance of the model in circuit simulation. Inductive loads can simulate the load characteristics of GaN HEMT devices in actual operation and accurately and clearly demonstrate the impact of the load on switch current waveform and voltage changes as shown in Figure 5. Thermal imaging of transistor temperature distribution is shown in Figure 6,which can be obtained by infrared spectroscopy.

## 3. Experimental Data Processing

### 3.1. Noise Filtering

In the switch current testing of GaN HEMTs, the measured signal may contain high-frequency noise due to the influence of experimental instruments, device factors, etc. Neural networks are highly sensitive to fluctuations in data sequences, and in order to effectively capture numerical patterns that reflect physical properties, it is necessary to denoise the test waveforms. These noises can have an impact on the characteristic analysis of GaN HEMT switching current testing and even affect model training and prediction. Therefore, before formally analyzing the characteristics of switch current, we need to perform noise filtering on the measured data. By filtering the data, the data quality can be improved, providing more reliable data for subsequent feature extraction, model training, and prediction.

The noise filtering method used in this article is simple moving average filtering, which creates a definable-sized rolling window that slides point by point on the data sequence. By calculating the average of adjacent data points, it can effectively suppress high-frequency noise in the signal. For sequences with large data fluctuations, moving average filtering can reduce data fluctuations, make the data smoother, highlight the long-term characteristics of the data, and facilitate subsequent analysis and processing of the data, such as calculating differences [22,23].

Assuming the input signal is x [n], the window size is N, and the output signal is y[n], the formula is as follows. Impact of moving average filtering on key waveform characteristics is shown in Table 1. Gate voltage data after filtering and filtered leakage voltage data are shown in Figure 7 and Figure 8, respectively.(1)$$y[n]=\frac{1}{N}\sum_{k=0}^{N-1}x[n-k]$$

### 3.2. Normalization Processing

The core step of data preprocessing in data normalization processing is to eliminate dimensional differences between different types of data, especially in the field of neural networks, which can improve the generalization ability and convergence speed of neural network models. In the dataset of this study, it is divided into time (s), voltage (V), and current (A). The numerical value range differences between different features are too large, and these differences in feature value ranges will affect the neural network model. If the dataset is directly imported into the neural network model for training, it will cause the weight of the neural network model to be imbalanced, thus failing to achieve ideal model training and prediction results.

Normalization of data can unify data with different features onto a relatively consistent scale, thereby eliminating dimensional differences between different features, making data distribution more concentrated, reducing the training time of neural network models, and improving the training efficiency and performance of neural network models. In this study, Min–Max Normalization was used, which can proportionally map the raw data to the [0, 1] interval through linear mapping, preserving the original shape of the data while eliminating the dimensional differences between different features. Its core formula is as follows:(2)$$x_{norm}=\frac{x-x_{min}}{x_{max}-x_{min}}$$
*x* is a numerical value in the original data, $x_{min}$ is the minimum value in the original dataset, $x_{max}$ is the maximum value in the original dataset, and $x_{norm}$ is the result of data normalization processing. If mapping to other intervals is needed, such as the [−1, 1] interval, Formula (2) can be adjusted:(3)$$x_{norm}=\frac{x-x_{min}}{x_{max}-x_{min}}\cdot (b-a)+a$$
where *a* and *b* are the upper and lower limits of the target interval, for example, *a* = −1, *b* = 1.

## 4. GaN HEMT Switching Current Model

### 4.1. Construction of Neural Network Models

In the construction of neural network models, we first need to define inputs and outputs. In this paper, the input data is the time series data of $v_{gs}$ (gate voltage) and $v_{ds}$ (drain voltage) of GaN HEMT at room temperature (T = 300 K). These input data cover the time dimension information and key switch characteristic parameters of GaN HEMT devices during operation. $v_{gs}$ controls the conduction and cutoff of GaN HEMT devices, while $v_{ds}$ reflects the voltage between the drain and gate.

Through this study, the aim is to output a predicted dynamic waveform of $I_{ds}$ (leakage source current), which directly reflects the working state of GaN HEMT devices. Because its dynamic current waveform reflects a lot of switching characteristic information, accurately predicting the switching current waveform of GaN HEMT devices is of great significance for understanding the performance of GaN HEMT devices and optimizing circuits.

Compared with traditional neural network models, this paper applies a hybrid network architecture of convolutional neural network and long short-term memory network (CNN-LSTM). For the CNN layer, one-dimensional convolutional neural networks (1D-CNN) are mainly used to extract local transient features. In the analysis of the switching characteristics of GaN HEMT devices, some transient phenomena, such as voltage overshoot, may occur. The one-dimensional convolutional neural network can automatically learn and extract local transient features of time series data by sliding convolution operations on the time series data through its convolution kernel, making these local transient features present a specific form in the local time window.

In a one-dimensional convolutional neural network, 32 filters are used with a kernel size of 1, which is equivalent to independent processing for each time step. The activation function used is relu, which can introduce nonlinear mapping to improve the feature extraction ability of local data. Then, the BatchNormalization layer is used to normalize the output data after passing through the convolutional layer, reducing the sensitivity of the model to the original data, accelerating the convergence speed of model training, and improving the stability of the model. Finally, the MaxPooling1D layer is applied to sample the output of the convolutional layer, reducing the dimensionality of the data while preserving its salient features.

For the LSTM layer, a stacked structure of dual LSTM layers is adopted. In the structure of the first LSTM layer, there are 64 units. The input source of the first LSTM layer is the output of the CNN layer. For each time step t, $h_{t}$ is calculated, and the outputs of all time steps are passed, returning the output of the entire sequence. The input of the LSTM layer in the second layer is the complete sequence output by the LSTM layer in the first layer. The LSTM layer in the second layer contains 32 units, discarding the intermediate time step states and only outputting the hidden state $h_{T}$ of the last time step:(4)$$h_{t}^{LSTM}=\mathrm{LSTM}(x_{t},h_{t-1})$$

The dual LSTM layer effectively models the transient response relationship between gate voltage, drain voltage, and current through the gate control mechanism and state transfer. The first LSTM layer extracts local feature details, while the second LSTM layer compresses global features. The combination of the two provides high-quality information density for subsequent fully connected layers.

Finally, feature fusion and output are performed. The 1D-CNN output layer and the dual LSTM layer output are flattened and concatenated. The two Dense layers include 128 and 64 neurons, respectively. The activation function uses the relu function, and the output of the fully connected layer is batch-normalized through two BatchNormalization layers. Two Dropout layers with dropout rates of 0.2 are used to prevent overfitting of the neural network model. Finally, a Dense layer containing only one neuron is used to output the predicted current value. 1D-CNN-LSTM flowchart is shown int Figure 9.

In the construction of this model, the loss function used is huber, and Adam optimizer is used with a learning rate of 1 × 10^−3^. This huber loss function can balance MSE and MAE and improve the robustness of the model, as shown in Equation (5). Meanwhile, root mean square error (RMSE) is used as the evaluation metric.(5)$$L_{\delta }(y,f(x))=\left\{\begin{array}{ll} \frac{1}{2}{\left(y-f(x)\right)}^{2} & \mathrm{if}\;|y-f(x)|\leq \delta \\ \delta \cdot |y-f(x)|-\frac{1}{2}\delta^{2} & \mathrm{otherwise} \end{array}\right.$$

### 4.2. Network Training

In the process of training the neural network models, we first need to partition the preprocessed data into appropriate datasets. For each temperature condition (300 K and 125 °C), the complete dataset consists of 200 independent switching waveforms, measured under the dynamic test conditions described in Section 2.2 (400 V DC bus voltage and a 0 V to +6 V gate swing). Each individual waveform comprises 1500 time-series data points, capturing the full transient event.

To ensure an unbiased evaluation of the model’s generalization ability, the dataset was partitioned on a per-waveform basis. The 200 waveforms were randomly divided into a training set and a testing set in an 8:2 ratio (i.e., 160 waveforms for training and 40 for testing). This partitioning strategy guarantees that the training and testing sets are completely independent, as no data points from a waveform used for testing are ever exposed to the model during the training process. Furthermore, 20% of the training set (32 waveforms) was set aside as a validation set. Unlike the test set, the validation set is used during the training process to monitor the model’s generalization ability, adjust hyperparameters, and implement early stopping to prevent overfitting. The final test set is used only once, after all training is complete, to provide a final, objective measure of the model’s performance.

Secondly, during the training process of the neural network model, when the neural network model is evaluated in the validation set, the performance of the neural network model no longer improves after multiple rounds. In order to avoid overfitting of the neural network model and ineffective training rounds and to save training time and computational consumption, this paper adopts an early stopping mechanism (Earlystopping) in the model training process to monitor the training indicators of the neural network model. If the training indicators do not improve within the rounds set by the early stopping method, the neural network model training will stop and restore the weights and parameters of the model with the minimum loss in the validation set so as to achieve the optimal training effect of the model.

Then, during the training process of the neural network model, dynamic learning rate adjustment was also adopted. In the early stage of training, a higher learning rate was used to ensure the training and convergence speed of the model. In the later stage, when the performance indicators of the neural network model stagnate, the learning rate will be automatically reduced so that the neural network training model can be fine-tuned under the optimal performance indicators to prevent local situations from falling into the optimum, thus balancing the training and convergence speed of the neural network model. The cosine annealing formula is as follows:(6)$$\eta_{t}=\eta_{\min}+\frac{1}{2}(\eta_{\max}-\eta_{\min})\left(1+\cos\left(\frac{t}{T}\pi \right)\right)$$
where $\eta_{\min}$ = 1 × 10^−6^, $\eta_{\max}$ = 1 × 10^−3^, and Cycle T = 20.

In the process of neural network training, a combination of early stopping mechanism and dynamic learning rate adjustment is used to first adjust the model learning rate when the model indicators do not improve in the set rounds in order to prevent the model performance from falling into local optima. When the learning rate decreases to the minimum value, if the training improvement is not satisfactory, early stopping will be performed. The combination of the two greatly prevents overfitting during model training, improves the speed of model training and convergence, and is more conducive to the subsequent evaluation of neural network models.

### 4.3. Model Validation

Before evaluating the model’s performance, it is important to consider the sources of uncertainty that influence the final prediction accuracy. The total error, captured by metrics like RMSE and MAE, is a composite of: (1) measurement uncertainty from the experimental equipment, (2) data processing uncertainty introduced during the noise filtering stage, and (3) model approximation error inherent to the neural network’s function fitting capabilities.

To specifically quantify the impact of the data processing step, a sensitivity analysis was performed. The primary results in this paper were obtained using a moving average filter with a window size of N = 10. To test the model’s sensitivity, we re-processed the test dataset using two adjacent window sizes, N = 9 and N = 11. The final, trained model was then evaluated on these alternatively filtered datasets. The resulting mean absolute errors (MAE) were 0.1240 for N = 10, 0.1245 for N = 9, and 0.1236 for N = 11. The total variation in MAE across these tests was less than 0.5%, demonstrating that the model is robust to minor changes in the filtering parameters. This low sensitivity indicates that the uncertainty introduced by this step has a limited impact on the final prediction performance.

In the process of validating neural network models, the first step is to select an appropriate loss function in order to accurately predict the overall waveform of switch current. In this study, root mean square error (RMSE), mean absolute error (MAE), and coefficient of determination (R^2^) were selected to evaluate the degree of fit between the predicted waveform and the actual waveform. The formula is as follows:(7)$$\mathrm{RMSE}=\sqrt{\frac{1}{N}\sum_{i=1}^{N}{\left(y_{i}-{\hat{y}}_{i}\right)}^{2}}$$(8)$$\mathrm{MAE}=\frac{1}{N}\sum_{i=1}^{N}|y_{i}-{\hat{y}}_{i}|$$(9)$$R^{2}=1-\frac{\sum_{i=1}^{N}{\left(y_{i}-{\hat{y}}_{i}\right)}^{2}}{\sum_{i=1}^{N}{\left(y_{i}-\bar{y}\right)}^{2}}$$

Then, after the training of the neural network model is completed, the training loss curve and validation loss curve are drawn, and the convergence of the model through the loss curve is observed as shown in Figure 10. If the validation loss is much greater than the training loss, it indicates that the neural network model training may be overfitting. If the validation loss first decreases and then increases, it also indicates that the model training may be overfitting.

From the comparison chart of the training loss curve and the validation loss curve above, it can be seen that both the validation loss and training loss curves first decrease and then stabilize, and the training loss is always greater than the validation loss. This proves that the neural network model fits well.

Next, the curves of the true values and predicted values for the training set and validation set, respectively, are drawn in Figure 11 and Figure 12. By observing the coefficient of determination R2, we can observe the degree of fit between the true values and predicted values in the dataset. By observing the degree of fit between the true value and predicted value curves, we can directly see the accuracy of the model’s predictions.

Based on the curve between the true and predicted values of the above dataset, the coefficient of determination (R^2^) is used to evaluate the performance of the model on the training and testing sets. The closer R^2^ is to 1, the better the fit of the neural network model to the data, and the higher the fit between the predicted values and the true value curve. If the coefficient of determination R^2^ of the training set is much larger than that of the testing set, it indicates that the model may overfit, over-learn from the training data, and have poor generalization ability. By observing the above two graphs, the coefficient of determination R^2^ of the training set and the test set are very close, and both approach 1, indicating that the fitting degree of the training set and the test set models is good, and the predicted values are very close to the real values. This indicates that the model has demonstrated good performance and generalization ability in the task of predicting the switching current of GaN HEMT devices at room temperature. Training set R^2^: 0.9959, MAE: 0.1216, and testing set R^2^: 0.9955, MAE: 0.1240.

To evaluate the versatility and robustness of the proposed CNN-LSTM architecture under different operating conditions, a separate set of experiments was conducted at a high temperature of 125 °C. The loss curve at T = 125° is shown in Figure 13. It is crucial to clarify that this dataset was used to train a completely separate and independent model, following the exact same architecture and training procedure as the room-temperature model. The 125 °C dataset was partitioned into its own training and testing subsets and was not used to test the model trained on 300 K data. The goal of this exercise was to demonstrate the general applicability of our modeling methodology, rather than the cross-temperature generalization of a single trained model. The performance of this high-temperature model is presented below.

This neural network model has good fitting performance under both room temperature and high temperature conditions. The fitting effect of the training set under high-temperature conditions is shown in Figure 14. The fitting effect of the test set under high-temperature conditions is shown in Figure 15. The training set R^2^ is 0.9960, and MAE is 0.1150, and the test set R^2^ is 0.9939, and MAE is 0.1192 under high-temperature conditions. The average coefficient is close to 1, and it can adapt to various working conditions under different temperature conditions.

### 4.4. Comparative Analysis and Model Benchmarking

To further validate the effectiveness of the proposed CNN-LSTM architecture, we performed a comparative analysis against both established modeling paradigms and alternative neural network structures.

Qualitative Comparison with Physics-Based Models:

Physics-based models, such as the Angelov model, are powerful tools for device characterization, particularly for steady-state, frequency-domain applications like RF amplifiers. These models offer deep physical insight but typically require a complex and labor-intensive parameter extraction process from extensive measurements. Our proposed data-driven approach, in contrast, is specifically tailored for modeling time-domain switching transients, which are of primary interest in power electronics applications. The end-to-end learning workflow of our model automates the feature extraction process, making it highly efficient for capturing the complex, nonlinear dynamics of switching events without requiring deep a priori physical knowledge. Therefore, our method represents a different but complementary modeling paradigm optimized for transient analysis.

Quantitative Comparison with an LSTM-Only Baseline:

To provide direct quantitative evidence of our model’s superiority for this task, we benchmarked the proposed CNN-LSTM architecture against a strong deep learning baseline: an LSTM-Only model. This comparison is designed to specifically isolate and demonstrate the contribution of the convolutional layers. The LSTM-Only model was constructed by removing the 1D-CNN front-end from our architecture and was trained on the exact same pre-processed dataset under identical conditions. Table 2 shows the key hyperparameters for model training.

The results in Table 3 clearly demonstrate the significant advantage of the hybrid CNN-LSTM architecture. The proposed model achieved a 28.8% reduction in RMSE and a 34.0% reduction in MAE, compared to the LSTM-Only baseline. This empirical evidence strongly suggests that the 1D-CNN layers are crucial for automatically learning and extracting the sharp, local transient features (e.g., current overshoots and ringing) present in the switching waveform. The LSTM-Only model, while capable of capturing temporal sequences, struggles to represent these high-frequency details with the same level of fidelity. This benchmark quantitatively validates that the synergistic combination of CNN and LSTM is superior for accurately modeling the complex dynamics of GaN HEMT switching currents.

## 5. Conclusions

This study addressed the significant challenge of accurately modeling the complex, nonlinear switching current dynamics of gallium nitride (GaN) high-electron mobility transistors (HEMTs). We proposed and validated a hybrid deep learning architecture combining a one-dimensional convolutional neural network (1D-CNN) with a long short-term memory (LSTM) network. The core mechanism of this approach leverages the 1D-CNN to automatically extract local, high-frequency transient features from the input voltage waveforms, while the LSTM layers effectively capture the temporal dependencies and memory effects inherent in the switching process. The efficacy of this methodology was demonstrated through its high-fidelity prediction of the device’s switching current. The final model achieved excellent performance on unseen test data, with coefficients of determination (R^2^) of 0.9955 at room temperature (300 K) and 0.9939 at high temperature (125 °C), providing strong quantitative evidence for the architecture’s accuracy and robustness.

Despite these promising results, it is crucial to acknowledge the limitations of the current study to properly contextualize its contributions. Firstly, the scope of generalization is confined to discrete operating points; the models developed are temperature-specific and do not possess the capability for continuous prediction across a thermal gradient. A separate model must be trained for each temperature. Secondly, while the quantitative comparison against an LSTM-Only baseline effectively validated the hybrid architecture’s design, the study lacks a direct performance benchmark against established physics-based or empirical models, such as the Angelov model. Finally, the findings presented here are based on a single type of GaN HEMT device operating under a specific resistive load condition. The model’s performance on different device structures or under more complex dynamic loads, such as inductive loads, remains an open question.

These limitations directly inform a clear roadmap for future research. The immediate next step is to develop a single, unified model that incorporates temperature as a continuous input variable, enabling true thermal generalization. A comprehensive benchmarking study against both traditional physics-based models and other machine learning approaches is also essential to rigorously position this method within the existing landscape of device modeling. Finally, future work will focus on expanding the methodology to a wider range of GaN HEMT devices and validating its performance under more diverse and practical operating conditions, including dynamic inductive loads, to further enhance its applicability for real-world power electronics design.

## Figures and Tables

**Figure 1 micromachines-16-00915-f001:**
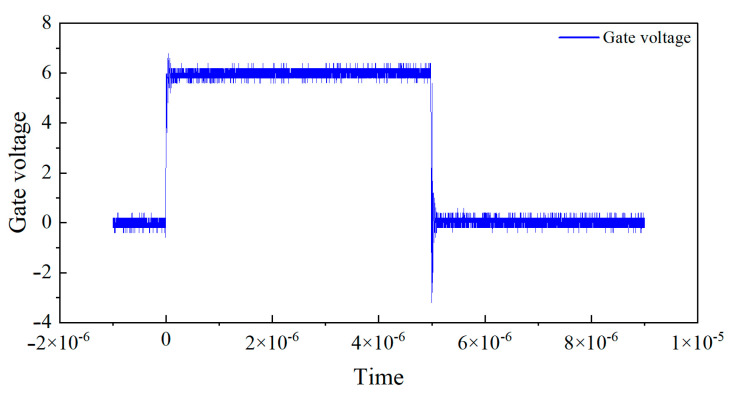
Pulse I–V static DC characteristics test gate voltage waveform.

**Figure 2 micromachines-16-00915-f002:**
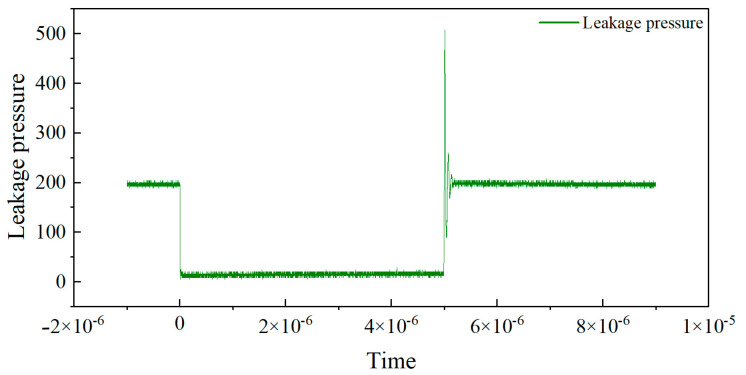
Leakage voltage waveform of pulse I–V static DC characteristic test.

**Figure 3 micromachines-16-00915-f003:**
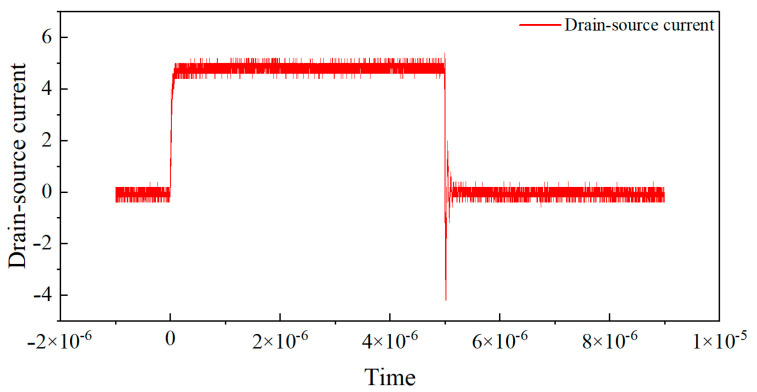
Leakage source current waveform of pulse I–V static DC characteristic test.

**Figure 4 micromachines-16-00915-f004:**
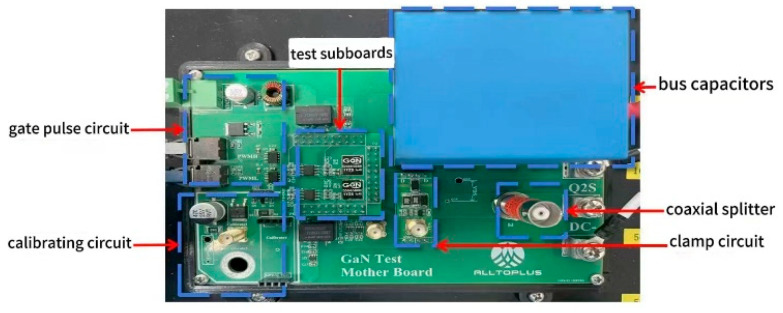
Layout of dual-pulse testing platform.

**Figure 5 micromachines-16-00915-f005:**
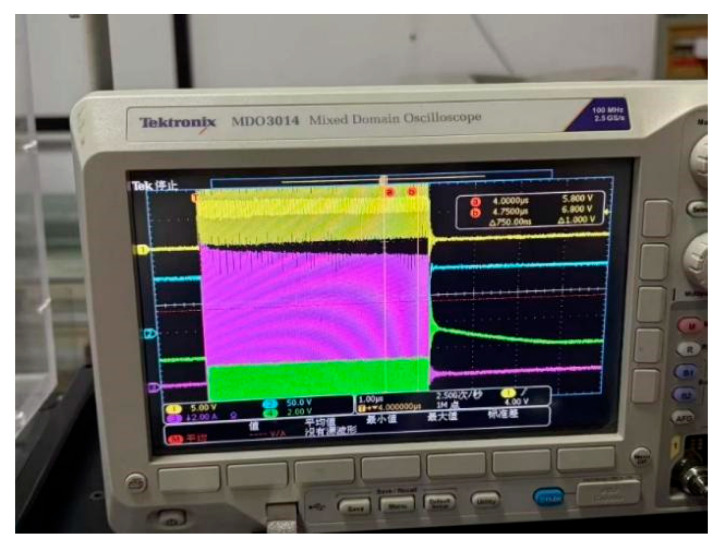
Oscilloscope screenshot.

**Figure 6 micromachines-16-00915-f006:**
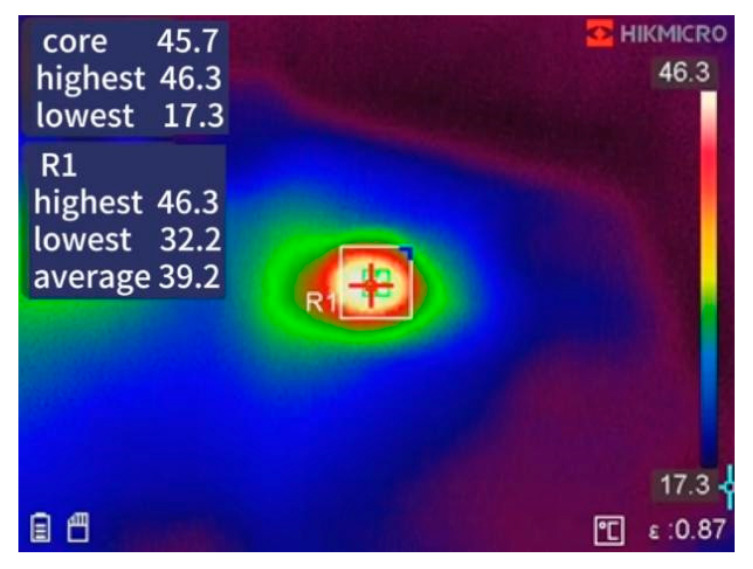
Thermal imaging of transistor temperature distribution.

**Figure 7 micromachines-16-00915-f007:**
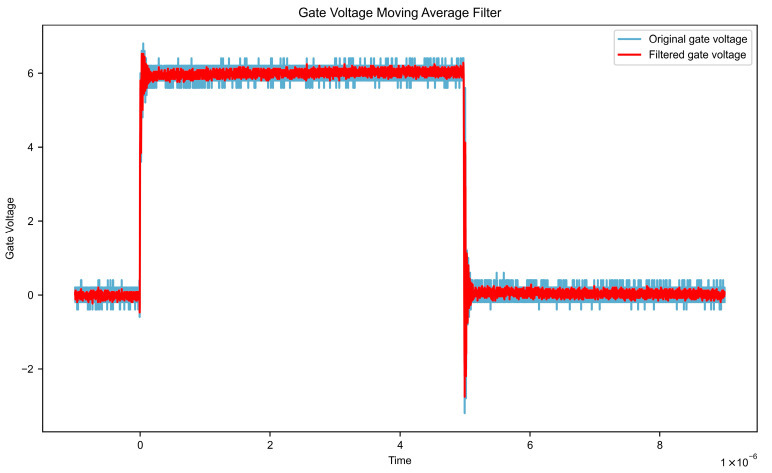
Gate voltage data after filtering.

**Figure 8 micromachines-16-00915-f008:**
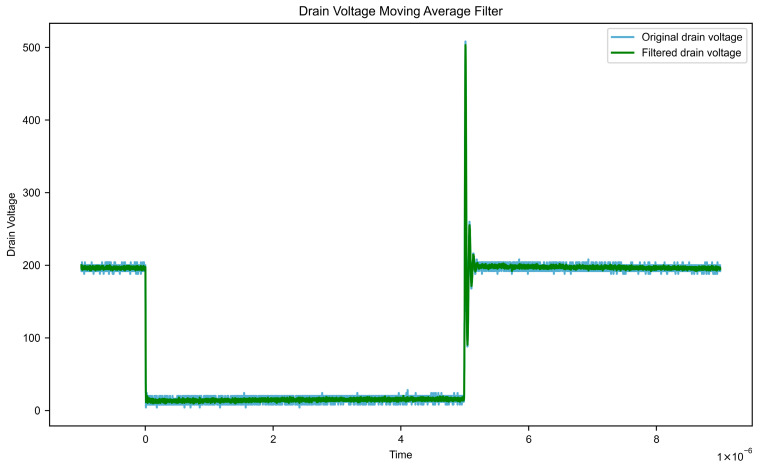
Filtered leakage voltage data.

**Figure 9 micromachines-16-00915-f009:**
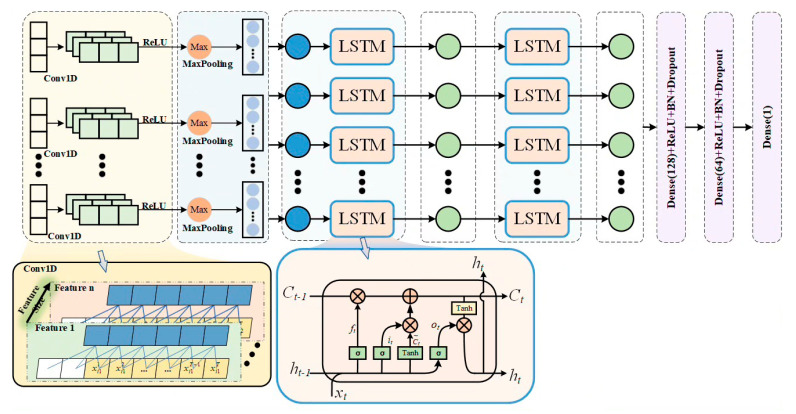
1D-CNN-LSTM flowchart.

**Figure 10 micromachines-16-00915-f010:**
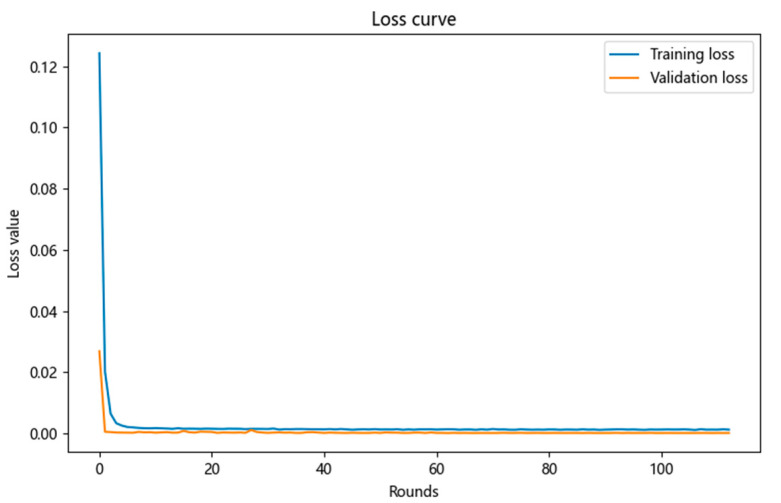
Training loss curve and validation loss curve.

**Figure 11 micromachines-16-00915-f011:**
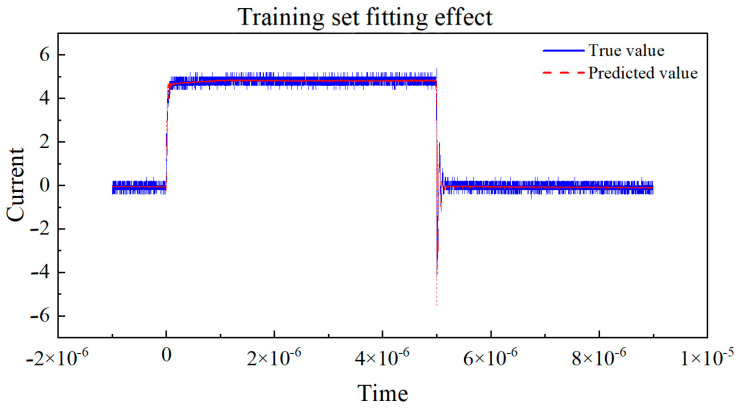
The true value vs. predicted value curve of the training set.

**Figure 12 micromachines-16-00915-f012:**
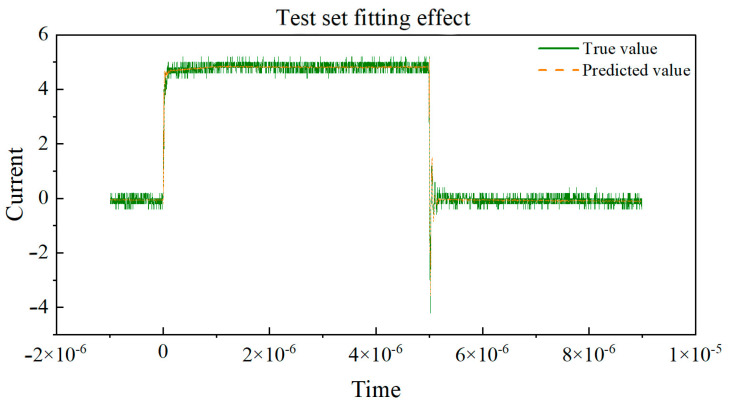
Real value vs. predicted value curve of the test set.

**Figure 13 micromachines-16-00915-f013:**
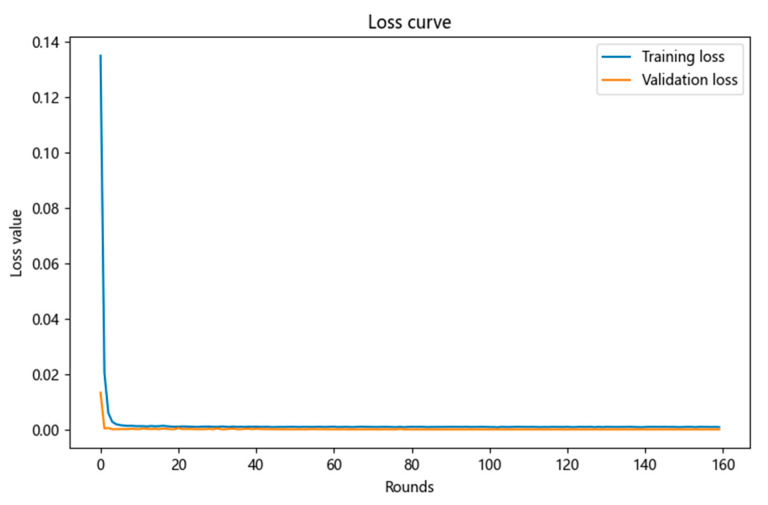
Loss curve at T = 125°.

**Figure 14 micromachines-16-00915-f014:**
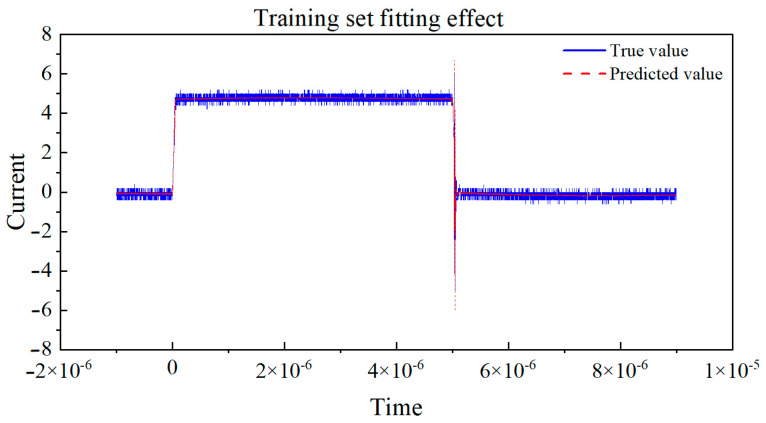
The fitting effect of the training set under high-temperature conditions.

**Figure 15 micromachines-16-00915-f015:**
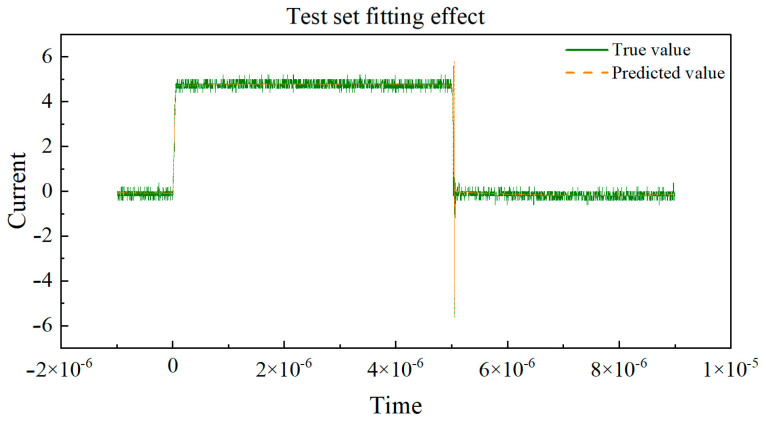
Fitting effect of the test set under high-temperature conditions.

**Table 1 micromachines-16-00915-t001:** Impact of moving average filtering on key waveform characteristics.

Waveform Characteristic	Raw Signal	Filtered Signal	Change
Rise Time/ns	31.586	27.396	−13.2%
Fall Time/ns	21.225	19.985	−5.8%
Overshoot Peak/V	537	509	−5.2%
Steady-State Noise/V	7.2	0.8	−88.9%

**Table 2 micromachines-16-00915-t002:** Key hyperparameters for model training.

Training Configuration		Training Configuration	
Optimizer	Adam	Batch Size	32
Loss Function	Huber	Number of Epochs	200
Initial Learning Rate	1 × 10^−3^	Monitored Metric	val_loss
Learning Rate Schedule	Cosine Annealing	Early Stopping Patience	15 epochs

**Table 3 micromachines-16-00915-t003:** Performance comparison of CNN-LSTM vs. LSTM-Only model (T = 300 K).

Model Architecture	RMSE	MAE	R^2^
LSTM-Only	0.215	0.188	0.9921
CNN-LSTM	0.153	0.124	0.9955

## Data Availability

The original contributions presented in the study are included in the article, further inquiries can be directed to the corresponding author.

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
