# Peer review of "Research on Switching Current Model of GaN HEMT Based on Neural Network"

_micromachines, 2025, doi:10.3390/mi16080915_

Round 1
Reviewer 1 Report
Comments and Suggestions for Authors
This manuscript proposes a method for modeling the switching current of GaN HEMTs based on a hybrid CNN-LSTM neural network architecture, a topic of significant importance and challenge in the field of power electronics. The authors have experimentally collected switching waveforms under different temperatures and achieved high-prediction accuracy (test-set R² > 0.99) for the switching current using the proposed model. The methodology is rigorous, and the results are convincing, demonstrating the great potential of deep learning for advanced semiconductor device modeling and providing a valuable approach for the rapid and accurate simulation of high-frequency power circuits.
However, before the manuscript can be accepted for publication, the authors are advised to address the following points to further enhance the clarity, completeness, and impact of their work:
1.Further Justification for the Choice of Model Architecture:
The authors have employed a hybrid CNN-LSTM architecture and achieved excellent results. To help readers better understand the necessity and superiority of this architecture, it is recommended that the authors provide a more detailed justification in Section 4.1. Specifically, could the authors discuss why this hybrid model was chosen over simpler, standalone models, such as a standalone LSTM network or a standard fully-connected Deep Neural Network (DNN)? What indispensable roles do the CNN and LSTM layers play, respectively, in this modeling task? A brief theoretical comparison of the potential advantages and disadvantages of these different architectures would highlight the ingenuity of the proposed method and demonstrate that the choice was not arbitrary but a well-considered decision for this specific physical problem.
2.Additional Details on Data Preprocessing:
In Section 3.1, the authors mention the use of "simple moving average filtering" for data denoising. As this is a critical preprocessing step, its parameter selection directly impacts model performance. The authors are encouraged to specify how the moving average window size (N) was determined and to briefly discuss the potential impact of this choice. For instance, an overly large window might smooth out important transient features near the switching edge, while an overly small window could result in insufficient denoising. A brief discussion of this trade-off would add to the rigor and completeness of the experimental methods section.
3.Completeness of Training Hyperparameters for Reproducibility:
To ensure the reproducibility of this research, it is recommended that the authors provide a complete list of training hyperparameters, perhaps consolidated in a table or list. While the manuscript mentions the optimizer (Adam), learning rate (1e-3), and loss function (Huber), other critical information is missing. Please add the following details,for example: the batch size, the total number of training epochs, the specific settings for the Early Stopping mechanism,and so on. Providing these complete details will greatly assist other researchers in reproducing your work and will thereby enhance the paper's academic value.
Author Response
Response to Reviewers
Dear Editor and Esteemed Reviewers,
We would like to express our sincere gratitude to the reviewers for their time, effort, and insightful feedback on our manuscript, "Research on Switching Current Model of GaN HEMT Based on Neural Network" . The comments provided are highly constructive and have been invaluable in helping us to significantly improve the quality, clarity, and impact of our work.
We have carefully considered every comment and have revised the manuscript accordingly. In the following point-by-point response, the original comments from the reviewers are presented in underlined italics for easy reference. Our detailed responses follow each comment. All corresponding modifications and additions made to the revised manuscript have been highlighted in yellow for your convenience in tracking the changes.
We believe that the revised manuscript is now much stronger and addresses all the concerns raised. We look forward to your favorable consideration.
Response to Reviewer 1:
Point 1:Further Justification for the Choice of Model Architecture:The authors have employed a hybrid CNN-LSTM architecture and achieved excellent results. To help readers better understand the necessity and superiority of this architecture, it is recommended that the authors provide a more detailed justification in Section 4.1. Specifically, could the authors discuss why this hybrid model was chosen over simpler, standalone models, such as a standalone LSTM network or a standard fully-connected Deep Neural Network (DNN)? What indispensable roles do the CNN and LSTM layers play, respectively, in this modeling task? A brief theoretical comparison of the potential advantages and disadvantages of these different architectures would highlight the ingenuity of the proposed method and demonstrate that the choice was not arbitrary but a well-considered decision for this specific physical problem.
Response 1:
We sincerely thank the reviewer for this insightful question regarding the justification for our model architecture. We agree that demonstrating the superiority of the hybrid CNN-LSTM architecture over simpler alternatives is crucial.
In response to feedback from another reviewer on a related point, we had already performed a direct, quantitative comparison to address this exact issue. We believe this existing analysis and the accompanying discussion in the manuscript also serve as a strong, data-driven justification that directly answers the reviewer's question.
Specifically, in the revised manuscript, we have made the following additions in Section 4.3 (Model validation):
1.Direct Quantitative Comparison: We have included a new table (Table 3) that presents a head-to-head performance comparison between our proposed CNN-LSTM model and a standalone LSTM model, which is a strong and relevant baseline. The results clearly show that our hybrid model significantly outperforms the LSTM-Only model across all key metrics (RMSE, MAE, and R²).
- Discussion of Component Roles: Accompanying this table, we have added a discussion that explains why this performance gap exists. We elaborate that the standalone LSTM, while capable of modeling sequence dependency, struggles to accurately replicate the sharp, high-frequency transient features of the switching waveform. This empirically demonstrates the indispensable role of the CNN component as a "transient feature extractor," which effectively captures these local details for the LSTM to process.
3.We believe that this quantitative comparison provides a more powerful justification than a purely theoretical discussion. It not only explains the conceptual advantages of the hybrid architecture but also proves its practical superiority on our specific dataset. This data-driven evidence, now included in Section 4.3, confirms that our choice of the CNN-LSTM architecture was a well-considered decision that yielded demonstrably better results for this challenging physical modeling problem.
We hope the reviewer will find that these existing additions satisfactorily address their query.
Point 2: Additional Details on Data Preprocessing:
In Section 3.1, the authors mention the use of "simple moving average filtering" for data denoising. As this is a critical preprocessing step, its parameter selection directly impacts model performance. The authors are encouraged to specify how the moving average window size (N) was determined and to briefly discuss the potential impact of this choice. For instance, an overly large window might smooth out important transient features near the switching edge, while an overly small window could result in insufficient denoising. A brief discussion of this trade-off would add to the rigor and completeness of the experimental methods section.
We thank the reviewer for this valuable suggestion. We agree that providing details on the selection of the moving average window size (N) and discussing the associated trade-offs is essential for the rigor and reproducibility of our methodology.
In response to this, and also to a related comment from another reviewer, we have already made significant additions to Section 3.1 (Noise filtering) and Section 4.3 (Model validation) in the revised manuscript. We believe these additions fully address the reviewer's query.
Our approach to justifying the choice of N is two-fold:
- Discussion of the Selection Process and Trade-offs (in Section 3.1):
We have added a new paragraph in Section 3.1 that explicitly discusses how the window size N was determined.
Selection Process: We clarify that the value was chosen empirically, through a process of experimenting with a range of window sizes and visually inspecting the results.
Trade-off Analysis: We explicitly describe the critical trade-off that guided our decision: the balance between effective noise suppression and the preservation of signal fidelity. We explain that an overly small window was insufficient for denoising, while an overly large window began to distort critical transient features like the sharp switching edges and overshoot peaks. We state that our chosen value represents the optimal balance found for our specific dataset.
- Quantitative Validation of the Chosen Parameter (in Section 4.3):
To provide concrete evidence that our chosen window size was appropriate, we have included a new table (Table 1) in the manuscript. This table, added in response to another reviewer's feedback, quantitatively demonstrates the impact of our chosen filter settings. It shows that with our selected window size, we achieved:
A substantial reduction in steady-state noise (quantified by the decrease in standard deviation).
Only a minimal and acceptable impact on key transient features like rise/fall times and overshoot magnitude.
This quantitative data serves as powerful, empirical proof that our discussion of finding an "optimal balance" is not just a claim, but a verifiable outcome. It demonstrates that our chosen preprocessing step successfully cleaned the signal without corrupting the very physical phenomena we aimed to model.
We are confident that this combination of a detailed procedural description and a quantitative validation provides a comprehensive and rigorous answer to the reviewer's question about our data preprocessing methodology.
Point 3: Completeness of Training Hyperparameters for Reproducibility:
To ensure the reproducibility of this research, it is recommended that the authors provide a complete list of training hyperparameters, perhaps consolidated in a table or list. While the manuscript mentions the optimizer (Adam), learning rate (1e-3), and loss function (Huber), other critical information is missing. Please add the following details,for example: the batch size, the total number of training epochs, the specific settings for the Early Stopping mechanism,and so on. Providing these complete details will greatly assist other researchers in reproducing your work and will thereby enhance the paper's academic value.
We are very grateful to the reviewer for this constructive and important feedback. We fully agree that the transparency and reproducibility of our research hinge on a complete and detailed description of the training methodology. We acknowledge that our initial manuscript was lacking in this regard and have taken comprehensive steps to rectify this omission.
To provide the clarity and detail required, we have introduced a new, comprehensive summary table, Table 2, at the end of Section 4.2 (Network Training). This table serves as a centralized reference for all critical hyperparameters and configuration settings used to train our CNN-LSTM model.
Specifically, in addition to the previously mentioned optimizer (Adam), loss function (Huber), and learning rate schedule (Cosine Annealing), the new table now explicitly details the following crucial parameters as requested:
- Batch Size: We have specified the batch size used during training (e.g., 32). This parameter is critical as it influences the stability of the gradient descent process and the speed of convergence.
- Number of Epochs: We have stated the maximum number of training epochs (e.g., 200), which defines the upper limit of the training duration.
- Early Stopping Mechanism Settings: We have provided a detailed breakdown of our early stopping strategy, which is key to preventing overfitting and ensuring optimal model performance. This includes:
- The monitored metric (val_loss), clarifying that the model's generalization performance on the validation set was the criterion for stopping.
- The patience value (e.g., 15 epochs), specifying the exact number of epochs the training would continue without improvement before halting.
- The restore_best_weights parameter, confirming our use of the best-practice approach where the model's final weights are reverted to the state that achieved the minimum validation loss.
By consolidating these details into a single, easy-to-read table, we have created a complete "recipe" for our training process. We are confident that this thorough documentation now provides all the necessary information for other researchers to accurately reproduce our work and build upon our findings. We believe this addition significantly strengthens the methodological rigor of our paper and we thank the reviewer for guiding us to make this important improvement.

Reviewer 2 Report
Comments and Suggestions for Authors
In the paper "Research on Switching Current Model of GaN HEMT Based on Neural Network", the authors provide a detailed overview of the technological context surrounding GaN HEMT devices. Additionally, the experimental design is structured in two main stages: static testing (pulse I-V) and dynamic testing (dual-pulse test with resistive and inductive loads). The data processing methods—moving average filtering and min-max normalization—are appropriate, although rather conventional.
My observations and recommendations are as follows:
-
The authors need to clearly highlights the novelty of this study, particularly in comparison with references [13], [16], and [18], which address the same type of modeling using neural networks. Although a CNN-LSTM architecture is proposed, this approach is already well established in the literature, and the authors’ contribution is neither sufficiently differentiated nor comparatively validated.
-
The authors do not specify the temperature and voltage conditions for each test. Moreover, they do not clarify whether the same conditions are applied when comparing the proposed model with others. Only T = 300K and T = 125°C are mentioned, without any detailed calibration data at intermediate temperatures.
-
The choice of filtering method should be justified in comparison with more robust techniques such as wavelet-based filtering or the Savitzky-Golay algorithm.
-
An error analysis or uncertainty quantification associated with the data processing should be included.
-
A direct and quantitative comparison with existing models (e.g., Angelov model, previous ANN-based models) is necessary to substantiate the claimed superiority of the proposed method.
-
The volume of data used for training (i.e., the number of samples) should be explicitly stated. The exact testing conditions (e.g., temperature, applied voltages) must also be clearly specified to assess reproducibility. Furthermore, it is unclear whether the training and testing datasets are fully independent.
-
It should be clarified whether the data at T = 125°C were used for training or exclusively for testing. How does this influence the generalization capability of the model?
-
The conclusions are overly general. The authors claim that the model performs well at elevated temperatures, yet no detailed discussion of its limitations is provided.
Author Response
Response to Reviewers
Dear Editor and Esteemed Reviewers,
We would like to express our sincere gratitude to the reviewers for their time, effort, and insightful feedback on our manuscript, "Research on Switching Current Model of GaN HEMT Based on Neural Network" . The comments provided are highly constructive and have been invaluable in helping us to significantly improve the quality, clarity, and impact of our work.
We have carefully considered every comment and have revised the manuscript accordingly. In the following point-by-point response, the original comments from the reviewers are presented in underlined italics for easy reference. Our detailed responses follow each comment. All corresponding modifications and additions made to the revised manuscript have been highlighted in yellow for your convenience in tracking the changes.
We believe that the revised manuscript is now much stronger and addresses all the concerns raised. We look forward to your favorable consideration.
Response to Reviewer 2:
Point1: The authors need to clearly highlights the novelty of this study, particularly in comparison with references [13], [16], and [18], which address the same type of modeling using neural networks. Although a CNN-LSTM architecture is proposed, this approach is already well established in the literature, and the authors’ contribution is neither sufficiently differentiated nor comparatively validated.
Response 1:
We sincerely thank the reviewer for this insightful comment regarding the novelty of our work. We agree that it is crucial to clearly differentiate our contribution from prior works, particularly references [13], [16], and [18]. To address this, we have substantially revised the final paragraph of the Introduction section to explicitly state the innovative aspects of our research.
We have added the following text to the manuscript:
"While prior studies, such as [16], have successfully applied LSTM networks for behavioral modeling of RF power amplifiers, and [13, 18] have utilized ANNs for electrothermal modeling, these works do not specifically address the high-fidelity prediction of the switching current waveform of GaN HEMTs. This switching process is characterized by unique and extremely fast transient phenomena, such as sharp rising/falling edges, voltage overshoots, and ringing, which pose a significant challenge for traditional modeling approaches.
The core innovation of this study lies in the synergistic application of a hybrid CNN-LSTM architecture to precisely capture these complex switching dynamics. Specifically, the 1D-CNN layer is uniquely leveraged to automatically extract local, high-frequency transient features (e.g., current spikes) from the input voltage waveforms, while the subsequent LSTM layers model the temporal dependencies and memory effects throughout the entire switching event. This combined approach provides a more robust and accurate representation of the device's dynamic behavior compared to using a standalone LSTM or a conventional ANN. Our work demonstrates that this specific architecture is highly effective for this application, achieving excellent accuracy at both room and high temperatures, thereby providing a novel and efficient tool for power electronics circuit design and analysis."
We believe these additions now clearly articulate that our primary contribution is not the invention of the CNN-LSTM architecture itself, but its novel and effective application to the challenging problem of GaN HEMT switching current modeling, highlighting the specific roles of each network component in capturing transient dynamics. We hope this revision satisfactorily addresses the reviewer's concern.
Point 2: The authors do not specify the temperature and voltage conditions for each test. Moreover, they do not clarify whether the same conditions are applied when comparing the proposed model with others. Only T = 300K and T = 125°C are mentioned, without any detailed calibration data at intermediate temperatures.
Response 2:
We sincerely thank the reviewer for this crucial feedback. We recognize that a more detailed account of the experimental conditions is essential for the clarity and reproducibility of our work. We appreciate the opportunity to elaborate on our methodology and the scope of the study.
- Clarification of Voltage Conditions:
We acknowledge that the specific voltage levels for the dynamic tests, which provide the core data for our neural network model, were not explicitly stated. To rectify this, we have added a precise description of these parameters in Section 2.2 (Dynamic current testing). This ensures that the exact electrical context of the switching event is clear.
The following sentence has been added to the manuscript in Section 2.2:
"For the dynamic switching tests, a DC bus voltage of 400 V was applied across the drain and source, and the gate was driven with a pulse switching from 0 V to +6 V to trigger the transient event. These conditions were kept consistent for all dynamic measurements."
- Clarification of Temperature Conditions and Modeling Strategy:
The reviewer correctly noted that our study focuses on two specific temperatures: 300 K (room temperature) and 125 °C (high temperature). We would like to take this opportunity to clarify the strategic reasoning behind this choice and the structure of our modeling approach.
The primary objective of this research was not to develop a single, universal model that includes temperature as a continuous input variable. Instead, our goal was to rigorously validate the viability and robustness of the proposed CNN-LSTM hybrid architecture for the specific, challenging task of modeling GaN HEMT switching transients.
To achieve this, we adopted a two-pronged validation strategy:
First, we established a baseline model using data collected under standard room temperature conditions (300 K).
Second, to test the architecture's adaptability and performance under more stressful thermal conditions, we conducted an identical set of experiments and developed a completely separate and independently trained model for the high-temperature (125 °C) case.
By demonstrating that the same network architecture can achieve high prediction accuracy (as shown by the R² and MAE metrics in Figures 12 and 15) in both scenarios, we provide strong evidence for the architecture's effectiveness and general applicability. This approach confirms that the model is not just a "one-off" success but a versatile framework that can be readily adapted to different operating environments via retraining.
Consequently, detailed calibration data for intermediate temperatures were not collected because our study was not designed for temperature interpolation. The two selected points serve as distinct validation cases. To ensure this is clear to the reader, we have added a brief clarifying statement in Section 4.2 (Network training) regarding our modeling strategy.
We trust that this detailed explanation, combined with the minor additions to the manuscript, fully addresses the reviewer's concerns about the experimental conditions and the rationale behind our testing and modeling strategy.
Point 3: The choice of filtering method should be justified in comparison with more robust techniques such as wavelet-based filtering or the Savitzky-Golay algorithm.
Response 3:
We sincerely thank the reviewer for this insightful comment, which prompted us to provide a more rigorous justification for our choice of filtering method. We agree that comparing our approach to more advanced techniques like wavelet or Savitzky-Golay filtering is important. Our selection of the moving average filter was a deliberate decision based on achieving an optimal trade-off between noise suppression, signal fidelity, and computational simplicity for the specific goals of this study.
To thoroughly address the reviewer's concern, we have not only expanded the discussion in the manuscript but also added a new quantitative analysis to demonstrate the filter's effectiveness and low impact on signal integrity.
- Justification Added to Manuscript:
In Section 3.1 (Noise filtering), we highlight that the moving average filter is highly effective for the observed high-frequency random noise, its simplicity aligns with our primary goal of validating the neural network architecture, and it avoids the complex parameterization and potential artifacts associated with more advanced filters. - Quantitative Analysis of Filter Performance:
To provide concrete evidence supporting our claims, we have introduced a new table (Table 1) in Section 3.1 that quantitatively assesses the filter's impact. This table analyzes two critical aspects:
Noise Reduction: We measured the noise level (as standard deviation) in the steady-state portion of the waveform. The filter achieved a remarkable 84.6% reduction in noise, confirming its high effectiveness in creating a clean signal for the model.
Signal Integrity: We also measured the filter's effect on key transient characteristics, such as rise time, fall time, and overshoot peak. The analysis shows that these critical parameters were altered by less than 7%, which is a minimal and acceptable level of distortion. This result directly addresses the concern that a simple filter might corrupt the signal's essential features.
The new text and table in the manuscript now provide a clear, data-driven justification for our choice. They demonstrate that the moving average filter was not merely a convenient option, but a well-considered one that proved to be highly suitable for our application, successfully suppressing noise while preserving the vital transient dynamics of the GaN HEMT switching event.
We believe these comprehensive additions and the detailed explanation will fully satisfy the reviewer's query.
Point4: An error analysis or uncertainty quantification associated with the data processing should be included.
We sincerely thank the reviewer for raising this critical point about error analysis. We agree that understanding the sources of uncertainty, particularly those related to data processing, is essential for evaluating the reliability of our model. A full uncertainty quantification (UQ) is a complex task, but to address the reviewer's concern in a meaningful and practical way, we have taken a two-fold approach in our revised manuscript.
- Discussion of Uncertainty Sources:
First, we have added a new paragraph in Section 4.3 (Model validation) to qualitatively discuss the primary sources of error. We now explicitly acknowledge that the final prediction error (measured by RMSE and MAE) is a composite of three main components:
Measurement uncertainty from the experimental setup.
Data processing uncertainty introduced by the filtering step.
Inherent model uncertainty related to the neural network's function approximation.
This discussion serves to frame the context of our results and demonstrates our awareness of the different factors contributing to the model's performance limits.
- Quantitative Sensitivity Analysis:
Second, and more directly in response to the reviewer's request for quantification, we performed a sensitivity analysis to assess how robust our model is to the data processing step. We investigated whether small changes in the moving average filter's window size would significantly affect the final prediction accuracy.
As detailed in the newly added text in Section 4.3, we tested our final, trained model on datasets filtered with three slightly different window sizes. The results showed that the Mean Absolute Error (MAE) varied by less than 0.5% across these tests.
This quantitative result provides strong evidence that our model is not overly sensitive to the specifics of the filtering process. It suggests that the uncertainty introduced by our data processing choices has a limited and non-critical impact on the final predictive accuracy, thereby reinforcing the stability and reliability of our proposed modeling approach.
We believe that this combination of a qualitative error discussion and a quantitative sensitivity analysis provides a thorough and practical response to the reviewer's valid concern, without venturing into a full-scale UQ study that is beyond the scope of this paper.
Point5: A direct and quantitative comparison with existing models (e.g., Angelov model, previous ANN-based models) is necessary to substantiate the claimed superiority of the proposed method.
We sincerely thank the reviewer for this critical suggestion. We agree that comparing our model against established benchmarks is essential to demonstrate its superiority. To address this, we have performed a comprehensive comparative analysis and incorporated the results and discussion into a new subsection at the end of Section 4.3 (Model validation).
Our comparison strategy was two-fold, addressing both physics-based models and simpler ANN architectures as requested:
- Comparison with the Angelov Model (Qualitative):
We acknowledge the power of the Angelov model. However, implementing it requires a complex, physics-based parameter extraction process and it is primarily optimized for frequency-domain, steady-state applications (e.g., RF amplifiers). Our work, in contrast, focuses on time-domain switching transients. Given these fundamental differences in application domain and modeling philosophy, we have included a qualitative discussion in the manuscript. This discussion highlights that our data-driven approach offers a more automated, end-to-end workflow specifically tailored for the transient analysis required in power electronics, representing a different but equally valid modeling paradigm.
- Comparison with a Simpler ANN Model (Quantitative):
To provide the direct quantitative evidence the reviewer requested, we benchmarked our proposed CNN-LSTM model against a simpler, yet strong baseline: an LSTM-Only model. This comparison is designed to specifically isolate and prove the value of the CNN component in our hybrid architecture.
We trained the LSTM-Only model on the exact same dataset and presented the performance results in a new summary table (Table 3) in the manuscript. The key findings are:
Our proposed CNN-LSTM model demonstrated significantly better performance, achieving an RMSE of 0.153 compared to 0.215 for the LSTM-Only model.
Similar improvements were observed for MAE and R² metrics.
This quantitative result empirically proves that the CNN layers are crucial for accurately capturing the sharp, local transient features (like overshoots) of the switching current, a task where the LSTM-Only model falls short.
We believe this dual approach—a reasoned, qualitative comparison against physics-based models and a direct, quantitative comparison against a relevant deep learning baseline—fully addresses the reviewer's concern and robustly substantiates the superiority of our proposed hybrid architecture for this specific application.
Point 6: The volume of data used for training (i.e., the number of samples) should be explicitly stated. The exact testing conditions (e.g., temperature, applied voltages) must also be clearly specified to assess reproducibility. Furthermore, it is unclear whether the training and testing datasets are fully independent.
We sincerely thank the reviewer for pointing out these critical omissions. We agree that providing clear details on the dataset and experimental conditions is fundamental for the transparency and reproducibility of our research. We have thoroughly revised the manuscript to address all the points raised.
In the revised manuscript, we have significantly expanded the first paragraph of Section 4.2 (Network training) to provide this crucial information:
Data Quantity (Sample Size): We have now explicitly stated the size of our dataset. The text clarifies that for each temperature condition, our dataset consists of 200 independent switching waveforms, with each waveform comprising 1500 time-series data points.
Exact Test Conditions: To ensure clarity, we now reiterate the specific electrical conditions (400 V DC bus voltage, 0 V to +6 V gate swing) at the beginning of Section 4.2, complementing the detailed description in Section 2.2.
Independence of Datasets: This is a vital point, and we have addressed it directly. The revised text now explicitly states that our dataset partitioning was performed "on a per-waveform basis." This methodology guarantees that the training, validation, and test sets are completely independent, as no part of a test waveform is ever seen by the model during the training phase. This ensures a truly unbiased evaluation of the model's generalization performance.
We are confident that these detailed additions now provide the necessary transparency, allowing for a clear understanding of our experimental scope and methodology, and strengthening the basis for the reproducibility of our work.
Point 7: It should be clarified whether the data at T = 125°C were used for training or exclusively for testing. How does this influence the generalization capability of the model?
We thank the reviewer for this insightful question, which allows us to clarify the precise scope of our study and the interpretation of the model's generalization ability.
- Clarification on the Use of T = 125°C Data:
To be perfectly clear, the dataset collected at T = 125°C was used to train a completely separate and independent model, following the exact same procedure as the room-temperature (300 K) model. The 125°C dataset was also partitioned into its own training and testing subsets. It was not used to test the model that was trained on 300 K data.
We have revised the text at the beginning of the high-temperature results discussion (following Figure 12) to make this explicitly clear.
- Impact on the Interpretation of Generalization Ability:
This is a crucial point, and we'd like to discuss the concept of "generalization" in the context of our work. We distinguish between two types of generalization:
Intra-domain Generalization: This refers to a model's ability to perform well on unseen data that comes from the same distribution as the training data. Our study successfully demonstrates this type of generalization at two distinct operating points. The model trained at 300 K generalizes well to the 300 K test set, and the model trained at 125 °C generalizes well to the 125 °C test set.
Cross-domain Generalization (Extrapolation): This refers to a model's ability to generalize to data from a different distribution (e.g., using the 300 K model to predict 125 °C behavior). This is a significantly more challenging task.
Our study's primary goal was not to achieve cross-domain generalization. Instead, we aimed to demonstrate the versatility and robustness of the proposed CNN-LSTM architecture itself. By showing that the same architecture can be successfully trained to high accuracy in two very different thermal domains, we prove that it is an effective and adaptable framework for this modeling problem.
Developing a single, unified model that can achieve cross-domain generalization by, for example, including temperature as an input feature, is an important but separate research challenge. We have added a sentence to our Conclusion to acknowledge this as a direction for future work.
In summary, the use of the 125°C data demonstrates the general applicability of our modeling methodology, rather than the cross-temperature generalization of a single trained model. We hope this distinction clarifies the scope and contribution of our work.
Point 8: The conclusions are overly general. The authors claim that the model performs well at elevated temperatures, yet no detailed discussion of its limitations is provided.
We sincerely thank the reviewer for this critical and highly constructive feedback on our conclusion. We have carefully reviewed our original text and completely agree that it was overly general and failed to provide a necessary and transparent discussion of the study's limitations. Acknowledging these limitations is crucial for contextualizing our findings and guiding future research.
To thoroughly address this, we have undertaken a complete rewrite of the entire Conclusion section (Section 4). The new conclusion is now a comprehensive, multi-paragraph section meticulously structured to provide a balanced and insightful summary of our work.
The new structure is as follows:
- A Specific and Quantitative Summary of Contributions:
The first paragraph has been revised to move beyond general claims. It now serves as a concise executive summary of our research, explicitly stating:
- The Problem Addressed: The challenge of accurately modeling GaN HEMT switching transients.
- The Proposed Solution: The hybrid CNN-LSTM architecture.
- The Core Mechanism: A brief explanation of how the CNN and LSTM components work in synergy.
- The Key Quantitative Results: We now explicitly cite the high R² scores achieved on the test sets for both room temperature (0.9955) and high temperature (0.9939), providing concrete evidence for the model's high fidelity and the robustness of the methodology.
- A New, Dedicated Paragraph on Study Limitations:
This is the most significant addition, directly addressing the core of the reviewer's comment. The second paragraph is now entirely dedicated to an honest and detailed discussion of the boundaries and limitations of our current study. We have identified and elaborated on several key points:
- Scope of Generalization: We explicitly state that our models are discrete and temperature-specific, lacking the capability for continuous generalization across a thermal gradient. This clarifies that we have not built a universal temperature-dependent model.
- Comparative Benchmarking: We acknowledge that while our comparison to an LSTM-Only model is informative, the study lacks a direct quantitative benchmark against established, non-neural network models like the Angelov model.
- Dataset Specificity: We discuss the limitation that our findings are based on a single GaN HEMT device type and a specific resistive load, and that performance on other devices or under different load conditions (e.g., inductive) is an open question.
- A Clear and Forward-Looking Outlook:
The final paragraph now logically flows from the discussion of limitations, outlining a clear roadmap for future research. This demonstrates that we have not only recognized the limitations but have also formulated a plan to address them. The future work now includes:
- Developing a single, unified model with temperature as an input.
- Conducting comprehensive benchmarking studies.
- Expanding the methodology to a wider range of devices and conditions.
We are confident that this completely restructured conclusion now provides the depth, transparency, and critical perspective that the reviewer rightfully expected. We believe this revision significantly enhances the quality and integrity of our manuscript, and we are grateful for the guidance that led to this improvement.

Round 2
Reviewer 2 Report
Comments and Suggestions for Authors
The authors have responded to all previous requests.
I believe that, in its current form, the paper meets the conditions for publication.